# Prognostic Factors for Iatrogenic Tracheal Rupture: A Single-Center Retrospective Cohort Study

**DOI:** 10.3390/jcm9020382

**Published:** 2020-02-01

**Authors:** Sebastian Krämer, Johannes Broschewitz, Holger Kirsten, Carolin Sell, Uwe Eichfeld, Manuel Florian Struck

**Affiliations:** 1Division of Thoracic Surgery, Department of Visceral, Transplant, Thoracic and Vascular Surgery, University Hospital Leipzig, 04103 Leipzig, Germany; sebastian.kraemer@medizin.uni-leipzig.de (S.K.); johannes.broschewitz@medizin.uni-leipzig.de (J.B.); carolin.sell@googlemail.com (C.S.); uwe.eichfeld@medizin.uni-leipzig.de (U.E.); 2Institute for Medical Informatics, Statistics and Epidemiology, Medical Faculty, University of Leipzig, 04103 Leipzig, Germany; holger.kirsten@imise.uni-leipzig.de; 3Department of Anesthesiology and Intensive Care Medicine, University Hospital Leipzig, 04103 Leipzig, Germany

**Keywords:** tracheal rupture, tracheobronchial, injury, laceration, iatrogenic, survival

## Abstract

Iatrogenic tracheal ruptures are rare but severe complications of medical interventions. The main goal of this study was to explore prognostic factors for all-cause mortality and rupture-related (adjusted) mortality. We retrospectively analyzed patients admitted to an academic referral center over a 15-year period (2004–2018). Fifty-four patients met the inclusion criteria, of whom 36 patients underwent surgical repair and 18 patients were treated conservatively. In a 90-day follow-up, the all-cause mortality was 50%, while the adjusted mortality was 13%. Rupture length was identified as a predictor for all-cause mortality (area under the curve, 0.84; 95% confidence interval (CI) 0.74–0.94) with a cutoff rupture length of 4.5 cm (sensitivity, 0.70; specificity, 0.81). Multivariate analysis confirmed rupture length as a prognostic factor for all-cause mortality (adjusted hazard ratio (HR) 1.5; 95% CI 1.2–1.9; *p* = 0.001), but not for adjusted mortality (HR 1.5; 95% CI 0.97–2.3; *p* = 0.068), while mediastinitis predicted adjusted mortality (HR 5.8; 95% CI 1.1–31.7; *p* = 0.042), but not all-cause mortality (HR 1.6; 95% CI 0.7–3.5; *p* = 0.243). The extent of iatrogenic tracheal rupture and mediastinitis might be relevant prognostic factors for all-cause mortality and adjusted mortality, respectively.

## 1. Introduction

Iatrogenic tracheal ruptures are rare but severe complications of medical interventions. Their incidence is reported to be approximately 0.005% in single-lumen intubations, 0.05–0.19% in double-lumen intubations, and up to 1% in percutaneous dilation tracheotomies [1,2,3,4,5,6,7,8,9,10,11,12,13,14]. The risk factors are unspecific and not applicable for identifying high-risk patients, given the high numbers of procedures that are performed daily. Due to different causative events, which are each associated with various risk factors (i.e., skills and training of the operator, anatomy and morbidity of the patient), clear prognostic determinants are difficult to identify. There is a common agreement that prognosis is widely dependent on the severity of underlying diseases, but studies focusing on outcome prediction are scarce [1,2,3,4,5,6,7,8,9,10,11,12,13,14]. In the literature, predictive power related to outcomes has been observed for delayed diagnosis and therapy in general, surgical or conservative treatment in particular, and occurrence of mediastinitis, sepsis, and suture insufficiency [1,2,3,4,5,6,7,8,9,10,11,12,13,14].

Recently, we reviewed the records of our patients requiring thoracotomy for surgical treatment of an iatrogenic tracheal rupture for information regarding their anesthetic management and perioperative complications [15]. Patients with no or minor perioperative complications appeared to have significantly lower rupture-related (adjusted) and 30-day mortality than patients who had major complications. Our results suggested that appropriate interdisciplinary communication of individual preinjury morbidity and one-lung ventilation back-up strategies to provide the best possible conditions for surgery might help avoid major complications. The rationale for the current study was to determine independent prognostic factors for all-cause mortality and adjusted mortality in consecutive patients with iatrogenic tracheal rupture undergoing surgical and nonsurgical treatment.

## 2. Materials and Methods

### 2.1. Study Patients, Design, and Definitions

We designed a retrospective, observational, single-center cohort study to analyze consecutive iatrogenic tracheal rupture patients for prognostic determinants. After approval of the Ethics Committee (IRB00001750, AZ: 484/18k), we reviewed the database of the University Hospital Leipzig for cases of iatrogenic tracheal rupture between 07/2004 and 12/2018. We defined iatrogenic tracheal rupture as an unintentional lesion of the posterior part of the trachea (*pars membranacea*) related to medical interventions. Patients with incomplete documentation and under 18 years of age were excluded. The University Hospital Leipzig is an academic medical center that provides advanced emergency care and interdisciplinary specialists for the treatment of iatrogenic tracheal ruptures.

### 2.2. General Management

Patients suspected of iatrogenic tracheal rupture underwent diagnostic confirmation using fiber-optic bronchoscopy and chest computed tomography (CT) and were transferred to the intensive care unit (ICU). All patients received broad-spectrum antibiotics for mediastinitis prevention. The decision for surgical or nonsurgical treatment was made by a team that included attending thoracic surgeons, intensivists, and pulmonologists, according to the patients’ condition, rupture characteristics, injury mechanism, and diagnostic results. All decisions were based on an individual approach of weighting risks and benefits of surgical repair or conservative treatment and did not follow a study protocol.

### 2.3. Data Collection

Patients’ data were obtained from paper-based and/or electronic charts. We analyzed patients regarding demographic data, including sex, age, body height, and weight, ICU stay prior to rupture, interfacility referral, causative procedure (single-lumen tracheal intubation, double-lumen intubation, tracheotomy-related, surgery-related), elective intubation, emergency intubation, rupture length, emphysema, mediastinitis, chronic obstructive pulmonary disease (COPD), steroid medication, surgical repair, nonsurgical treatment, length of stay in the intensive care unit (LOS ICU), ventilator days after verification of tracheal rupture and/or surgical repair, adjusted mortality, and all-cause hospital mortality.

### 2.4. Statistical Analysis 

The distribution of continuous variables was tested using the Shapiro–Wilk normality test. Data were expressed as the mean ± standard deviation (SD), number (*n*) or percentage. Continuous variables were compared between groups using an unpaired t-test. Categorical variables between groups were compared using Fisher’s exact test. The significant variables in the univariate analysis were selected for multivariate analysis. A receiver operating characteristics curve was applied to discriminate between rupture length and all-cause mortality. The area under the curve was calculated, and the cutoff rupture length was identified using the Youden index. The positive predictive value, negative predictive value, sensitivity, and specificity were calculated at the cutoff point. The time to mortality events was modeled using the Cox proportional forward hazards model. A Kaplan–Meier survival plot was calculated from baseline to the time of mortality events and compared using the log rank test. All tests were two-sided with a significance level at *p* < 0.05. Statistical analysis was performed using *R*.

## 3. Results

### 3.1. Patient Screening and Inclusion Process

Patients in the study cohort were identified using International Classification of Diseases (ICD) revision 10 coding. ICD-10 codes, S11.x and S27.x, were reviewed, and all patients with iatrogenic tracheal ruptures were included. Of the 83 patients who were initially identified, 29 were excluded for the following reasons: patients age <18 years, seven patients; incorrect or erroneous coding (no mention of any tracheal injury in the medical records), eight patients; duplicate or multiple coding, six patients; elective esophageal, tracheal, or tracheobronchial surgery, four patients; tracheal injury not iatrogenic (blunt or penetrating injury), three patients; and tracheal rupture due to caustic ingestion, one patient.

### 3.2. Baseline Characteristics of Patients

Fifty-four consecutive patients with iatrogenic tracheal rupture were included, of whom 40 were female and 14 were male, with a mean (SD) age of 66.9 (16.3) years (Table 1 and Appendix A). Nineteen patients (35%) were referred from other hospitals. The main causative event of tracheal rupture was single-lumen tracheal intubation (35 patients), followed by tracheotomy (percutaneous dilatation tracheotomy, surgical tracheotomy, or tracheal cannula reinsertion and bronchoscopy associated injuries, 13 patients) and surgery (esophageal surgery and otolaryngology surgery, six patients). Subcutaneous emphysema was present in 44 patients (81%) as the primary clinical symptom, while nine patients (19%) remained asymptomatic. Mediastinitis was present in 14 patients (26%). The mean (SD) rupture length was 4.3 (1.8) cm. The adjusted mortality was 13%, while the all-cause hospital mortality was 50%. Only ICU admission prior to rupture showed significant differences across causative events (tracheotomy-related patients had almost all been admitted to the ICU prior to rupture, compared with patients with intubation-related and surgery-related causes).

Tracheal intubation was performed in 94% (*n* = 33) of patients as an emergency measure, and of these, 34% (*n* = 12) were during cardiopulmonary resuscitation. A stylet was used in 80% (*n* = 28) of patients, while multiple laryngoscopy attempts were required in 20% (*n* = 7). Thirteen patients (37%) underwent prehospital intubation by an emergency medical service physician. The majority of intubation-related tracheal rupture patients were female (74%; *n* = 26), of whom 23 (88%) were aged over 50 years, and 77% (*n* = 20) were ≤165 cm tall. In five patients, intubation-related tracheal rupture remained undetected until percutaneous dilation tracheotomy and/or bronchoscopy 2, 6, 7 (two patients), and 16 days after tracheal intubation. All other patients were identified and treated within one day after rupture.

Among the whole study cohort, surgical repair was performed in 36 patients (67%), of whom 35 underwent right-side thoracotomy and one patient underwent an anterior cervical surgical approach. Eighteen patients underwent nonoperative treatment, either due to minor lesions and noncritical conditions (72%; *n* = 13) or due to best supportive care approaches in highly critical patients (28%; *n* = 5). Patients receiving surgical repair had higher rates of emphysema (*p* < 0.001), interfacility referral (*p* = 0.014), and COPD (*p* = 0.016) than patients without surgical repair (Table 2). Furthermore, the rupture length was comparable in patients with and without mediastinitis (*p* = 0.312).

### 3.3. Analysis of All-Cause and Adjusted Mortality

The mean follow-up period to mortality was 77.4 (86.2) days, while data of all patients were obtained at least until 90 days after diagnosis of iatrogenic tracheal rupture. Tracheal intubation, tracheotomy, and surgery as the main causes of tracheal rupture were comparably related to all-cause mortality-free survival (log-rank *p* = 0.366) and adjusted mortality-free survival (log-rank *p* = 0.271) (Figure 1).

All-cause and adjusted mortality were similar regarding emergency intubation compared with elective intubation (*p* = 0.244 and *p* = 1, respectively), stylet use compared with non-stylet use (*p* = 0.672 and *p* = 0.562, respectively), multiple attempts compared with first tube pass success (*p* = 0.672 and *p* = 0.171, respectively), and surgical repair compared with nonsurgical treatment (*p* = 0.773 and *p* = 1) (Table 2).

Univariate analysis revealed rupture length (hazard ratio (HR) 1.6; 95% confidence interval (CI) 1.3–2; *p* < 0.001), mediastinitis (HR 2.3; 95% CI 1–1.5; *p* = 0.035), and emphysema (HR 8.3; 95% CI 1.1–62; *p* = 0.038) as predictive parameters for all-cause mortality, while mediastinitis (HR 8.2; 95% CI 1.6–43; *p* = 0.012) and rupture length (HR 1.7; 95% CI 1.1–2.7; *p* = 0.019) had an effect on adjusted mortality (Table 3).

Multivariate analysis confirmed rupture length as a prognostic factor for all-cause mortality (HR 1.5; 95% CI 1.2–1.9; *p* = 0.001), which was observed with a comparable HR at trend-level significance for adjusted mortality (HR 1.5; 95% CI 0.97–2.3; *p* = 0.068), while mediastinitis predicted adjusted mortality (HR 5.8; 95% CI 1.1–31.7; *p* = 0.042) but not all-cause mortality (HR 1.6; 95% CI 0.7–3.5; *p* = 0.243) (Table 4).

### 3.4. Analysis of Rupture Length and Mortality

The receiver operating characteristics curve identified rupture length as an appropriate discriminator of all-cause mortality (Youden index 0.76) (Figure 2). We chose the cutoff length at the best Youden index, as this point provided the highest sum of sensitivity and specificity to predict all-cause mortality. The characteristics of patients with rupture sizes of <4.5 cm and ≥4.5 cm are provided in Table 5. The causative events were comparable in both groups (*p* = 1).

Kaplan–Meier survival curves at the 90-day follow-up comparing rupture sizes <4.5 cm and ≥4.5 cm also demonstrated significant differences for all-cause mortality-free survival (log-rank *p* < 0.001) and adjusted mortality-free survival (log-rank *p* = 0.002), respectively (Figure 3).

## 4. Discussion

### 4.1. Key Results and Interpretation

Iatrogenic tracheal rupture is a potentially life-threatening complication associated with high morbidity and poor overall survival, even in specialized centers. In our study, only every second patient survived this complication, while adjusted mortality reached 13%. These high rates reflect that most of our patients were critically ill or had undergone airway management measures due to acute emergency conditions. Our results are in line with other studies and our own previous results that preinjury morbidity and underlying disease play key roles in outcome and that adjusted mortality numbers are considerably lower compared with those of all-cause mortality [2,3,4,5,8,9,10,13,14].

Studies involving higher patient numbers or prospective randomized controlled trials of iatrogenic tracheal rupture are not available. Only a few studies have included cohorts with only one causative procedure (e.g., single- and/or double-lumen tracheal intubation-related, tracheotomy-related, or surgery-related procedures) [2,9,16], and most studies included different proportions of these iatrogenic causes [5,6,8,10,11,12,13,14]. Furthermore, some studies also present a variety of other, noniatrogenic causes, leading to tracheobronchial injuries [11,17].

Different causative procedures of iatrogenic tracheal rupture may be associated with different patterns of injuries. Tracheal intubations are the most common causes of iatrogenic tracheal ruptures, and patient-related risk factors may be short body height, female sex, old age, chronic obstructive pulmonary disease, and steroid use, which may be related to increased vulnerability of respiratory system tissue [2,5,6,7]. Operator-related factors may be a lack of procedural experience, emergency conditions, inappropriate use of stylets and large tube sizes, and cuff over-inflations [1,2,3]. Collectively, we found only moderate agreement (apart from emergency versus elective intubation) with these rather unspecific risk factors in our study. For tracheotomy-related ruptures and surgery-related ruptures, some of these risk factors may also be applicable, but detailed data are not available. In our study, all tracheotomy-related ruptures were opposite the tracheotomy site at the posterior wall of the trachea. Furthermore, the role of prior thoracic surgery on the severity of iatrogenic rupture and its outcome was difficult to assess and should be evaluated in future studies.

### 4.2. Prognostic Variables

Prognostic variables for all-cause and adjusted mortality in iatrogenic tracheal rupture are highly unspecific, as mentioned before. However, our multivariate analysis results suggest new independent predictors for all-cause mortality and adjusted mortality.

Rupture length was significantly associated with all-cause mortality, although it has previously not been identified for the prediction of any outcome [2]. Recently, Herrmann et al. published a large series of 64 patients with a hospital mortality of 15.6% (*n* = 10); of these, 30 patients with a rupture length of up to 4 cm and 24 patients with a rupture length exceeding 4 cm survived (*p* = 0.498) [14]. They concluded that there was no association between rupture length and mortality. There are several smaller studies in which non-survivors had slightly larger ruptures than survivors, namely, Leinung et al., formerly affiliated with our center, (3.8 cm *vs.* 3.5 cm), Sippel et al. (7.1 cm *vs.* 3.5 cm), and Lee (5 cm *vs.* 4 cm) [8,9,13]. However, Hofmann et al. and Deja et al. found even smaller mean rupture lengths in non-survivors compared to survivors (4.3 cm *vs.* 4.9 cm and 4.0 cm *vs.* 4.6 cm, respectively) [5,12]. These contradictory results raise several questions regarding the generalizability of our results, and further studies are needed to confirm them.

Apart from rupture length, the depth of the laceration is another injury characteristic that may have prognostic relevance [14,18]. Superficial injuries may have better healing potential than full-thickness injuries, which are associated with higher complication rates (i.e., obstruction of the tracheal lumen due to prolapsed mediastinal structures) [4,12,14,18]. However, in the present study, we could not obtain details regarding injury depths from all patients, which is certainly a limitation.

For adjusted mortality, mediastinitis was identified as an independent prognostic parameter. Acute mediastinitis is a severe infectious complication leading to sepsis and progressive multiorgan dysfunction [19]. Our results support previous studies in which mediastinitis was associated with mortality and suture insufficiency in univariate analyses [8,14]. Furthermore, mediastinitis is accepted as an indicator for surgical exploration after tracheal injury, and patients at high risk for mediastinitis usually receive broad-spectrum antibiotic administration [5,8,10,13,14,15,19,20]. Although the rupture lengths were similar in patients with and without mediastinitis, future studies should investigate whether rupture characteristics and patient conditions might influence the occurrence of mediastinitis.

### 4.3. Surgical and Non-Surgical Treatment

There is continuing controversy about surgical versus nonsurgical treatment of iatrogenic tracheal rupture [2,13,14]. In recent years, the number of publications favoring nonsurgical treatments has increased considerably [6,11,18,20,21]. Nonsurgical treatments include either strictly conservative treatment via frequent bronchoscopic evaluation, local fibrin-glue application, or the placement of stents [11,14,18,20,21,22]. In our study, nonsurgical patients were either too unstable for surgical intervention, due to critical underlying conditions, or healthy enough to undergo early tracheal tube removal and spontaneous breathing. Currently, we support interdisciplinary patient-centered treatment strategies that are in line with those suggested by other authors [5,8,10,12,13,14]. Nonsurgical management may be considered in healthy patients with spontaneous breathing, superficial ruptures, and low risk for developing mediastinitis and septic complications.

However, there are patients in whom the earliest possible surgical repair is unavoidable. Impairment of gas exchange caused by a lack of appropriate ventilation due to massive leakage volumes and high risk of mediastinitis and/or sepsis are strong indicators for surgical repair [5,8,9,10,11,13,14]. Surgical repair in ruptures localized in the cervical parts and upper thoracic parts of the trachea may be performed using anterior cervical approaches. Deeper ruptures reaching the tracheal bifurcation, the right main bronchus, and the proximal left bronchus usually require right thoracotomy under one-lung ventilation [2]. Left thoracotomy or median sternotomy are only required in complex and deep injuries to the left bronchus. In our study, surgical repair was almost entirely performed using a right thoracotomy, suturing, and suture reinforcement using pericardium, pleura, the thymus, or muscle tissue [15]. Only one patient received an anterior cervical approach for surgery. Other authors have published higher rates of this less invasive surgical access [7,10,13,14].

There is certainly a gray zone of patients who may benefit from both surgical and nonsurgical treatment, and each center has to determine the strengths and weaknesses for each individual patient, according to existing infrastructure, adjusting for local treatment protocols [13,14]. Future studies should further explore how alternative surgical approaches (i.e., endoscopy) may be transferred into broader clinical practice [22,23].

### 4.4. Limitations

The results of single-center retrospective cohort studies are not generalizable to other populations. They are unable to answer whether surgery or conservative treatment is superior. Our high preinjury morbidity and center-specific proportions of causative events should be kept in mind. Due to considerable variability in the underlying injury mechanisms and preinjury morbidity (e.g., rates of emergency intubations in critical patients versus elective intubations in healthy patients), the outcomes of available studies are difficult to compare. Long observational periods and small sample sizes further impair the generalizability of the results. Furthermore, we could not obtain data regarding injury depth and associated classifications. Although we present new data using multivariate tests, these results should be interpreted cautiously.

## 5. Conclusions

The extent of iatrogenic tracheal rupture and mediastinitis might be relevant prognostic factors for all-cause mortality and adjusted mortality, respectively. In clinical practice, these parameters could help to identify patients at high risk. To improve individual patient outcomes and for better comparability of study results, future study protocols should be strictly adjusted to causative events, patient morbidity, and treatment.

## Figures and Tables

**Figure 1 jcm-09-00382-f001:**
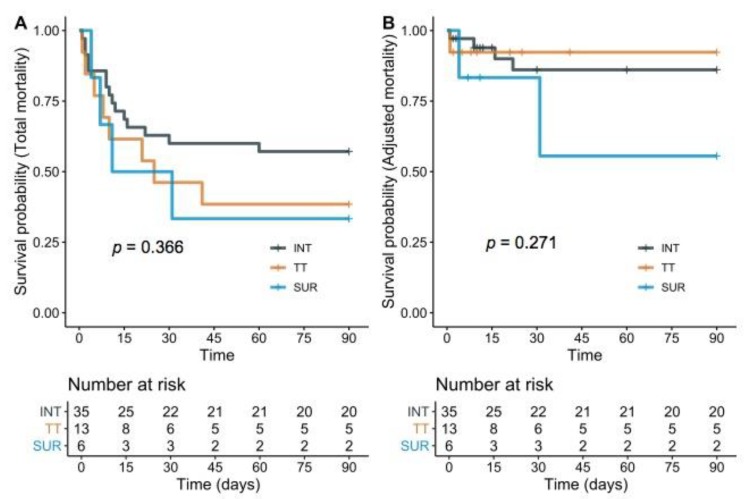
Kaplan–Meier survival probability related to causative events. Tracheal intubation-related (INT); tracheotomy-related (TT); surgery-related (SUR); *p*-value (*p*). Causes of iatrogenic tracheal rupture were comparable in all-cause mortality-free survival ((**A**); log-rank *p* = 0.366) and adjusted mortality-free survival ((**B**); log-rank *p* = 0.271).

**Figure 2 jcm-09-00382-f002:**
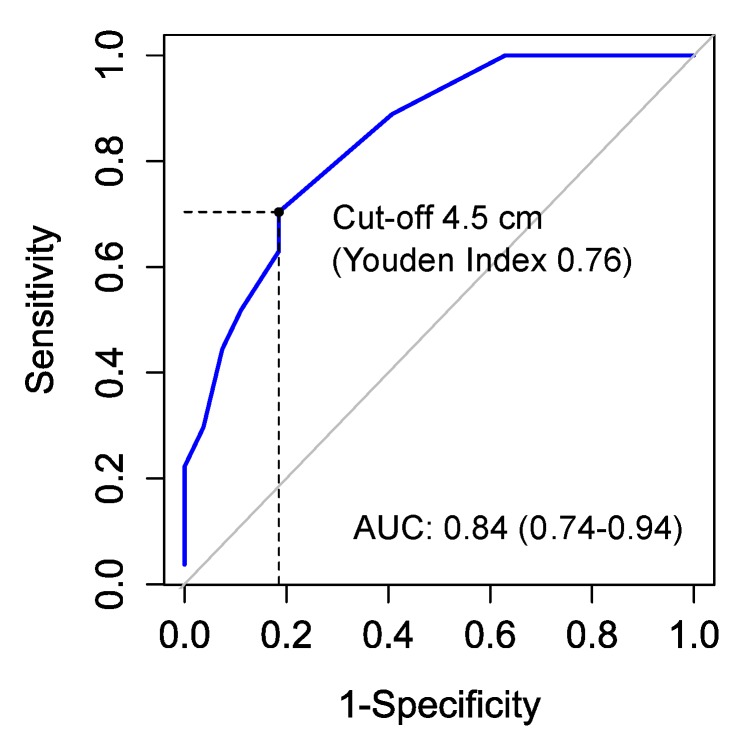
Receiver operating characteristics curve for rupture length and all-cause mortality. (Area under the curve (AUC), 0.84 (95% CI 0.74–94)). The cutoff length was identified as 4.5 cm with a sensitivity of 0.7 (95% CI 0.5–0.86) and a specificity of 0.81 (95% CI 0.62–0.94). The positive predictive value at a cutoff length of 4.5 cm was 0.79 (95% CI 0.58–0.91), while the negative predictive value at a cutoff length of 4.5 cm was 0.73 (95% CI 0.53–0.90).

**Figure 3 jcm-09-00382-f003:**
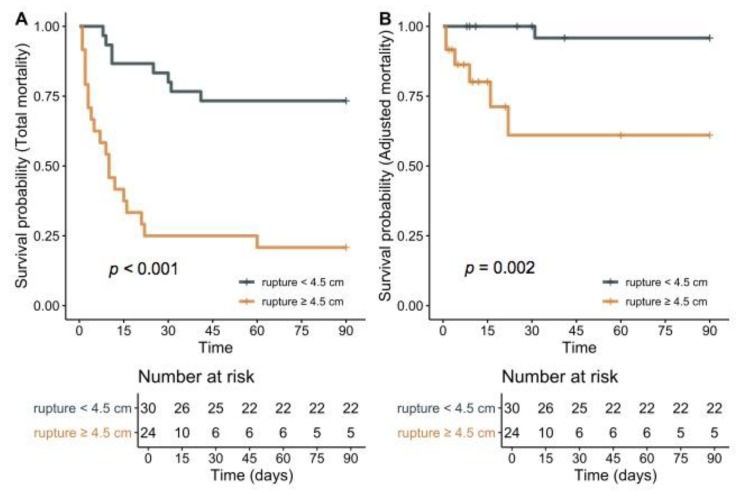
Kaplan–Meier curves in patients with rupture lengths of <4.5 cm vs. ≥4.5 cm for all-cause mortality-free survival ((**A**); log rank *p* < 0.001) and adjusted mortality-free survival ((**B**); log-rank *p* = 0.002).

**Table 1 jcm-09-00382-t001:** Comparison of characteristics between different causative events.

Parameter	Overall	INT	TT	SUR	*p*
Patients; *n*	54	35	13	6	
Female; *n* (%)	40 (74.1)	26 (74.3)	10 (76.9)	4 (66.7)	0.903
Age, years; mean (SD)	66.9 (16.3)	68.5 (17.7)	64.6 (11.9)	62.3 (16.7)	0.261
Weight, kg; mean (SD)	76.1 (18.8)	74.9 (16.7)	79.2 (24.0)	76.0 (20.9)	0.795
Height, kg; mean (SD)	164.6 (6.7)	164.3 (6.7)	165.7 (7.6)	164.5 (5.7)	0.809
ICU prior rupture; *n* (%)	29 (53.7)	16 (45.7)	12 (92.3)	1 (16.7)	0.001
Interfacility referral; *n* (%)	19 (35.2)	13 (37.1)	5 (38.5)	1 (16.7)	0.761
Rupture length; mean (SD)	4.3 (1.8)	4.4 (2.0)	4.3 (1.7)	4.2 (1.4)	0.962
Emphysema; *n* (%)	44 (81.5)	27 (77.1)	11 (84.6)	6 (100.0)	0.592
Mediastinitis; *n* (%)	14 (25.9)	8 (22.9)	3 (23.1)	3 (50.0)	0.456
Steroids; *n* (%)	15 (27.8)	10 (28.6)	4 (30.8)	1 (16.7)	1
COPD; *n* (%)	18 (33.3)	12 (34.3)	4 (30.8)	2 (33.3)	1
Surgical repair; *n* (%)	36 (66.7)	20 (57.1)	10 (76.9)	6 (100.0)	0.089
Respirator days; mean (SD)	12.6 (11.8)	13.5 (12.5)	11.5 (10.5)	9.7 (11.3)	0.641
ICU days; mean (SD)	17.0 (14.3)	17.8 (15.3)	15.8 (13.0)	14.7 (12.0)	0.835
Adjusted mortality; *n* (%)	7 (13)	4 (11.4)	1 (7.7)	2 (33.3)	0.309
All-cause mortality; *n* (%)	27 (50)	15 (42.9)	8 (61.5)	4 (66.7)	0.415

Tracheal intubation-related (INT); tracheotomy-related (TT); surgery-related (SUR); intensive care unit (ICU); chronic obstructive pulmonary disease (COPD); *p*-value (*p*); number (*n*); standard deviation (SD). Only ICU admission prior to rupture showed significant differences across causative events (tracheotomy-related patients had almost entirely been admitted to the ICU prior to rupture, compared with patients of intubation-related and surgery-related causes).

**Table 2 jcm-09-00382-t002:** Comparison of characteristics between patients with and without surgical repair.

Parameter	No Surgical Repair	Surgical Repair	*p*
Patients; *n*	18	36	
Female; *n* (%)	14 (77.8)	26 (72.2)	0.751
Age, years; mean (SD)	71.3 (17.8)	64.7 (15.3)	0.068
Weight, kg; mean (SD)	72.4 (14.1)	77.9 (20.7)	0.295
Height, cm; mean (SD)	162.6 (6.6)	165.6 (6.6)	0.118
ICU prior rupture; *n* (%)	9 (50.0)	20 (55.6)	0.776
Interfacility referral; *n* (%)	2 (11.1)	17 (47.2)	0.014
Rupture length, cm; mean (SD)	3.7 (2.0)	4.7 (1.7)	0.078
Emphysema; *n* (%)	10 (55.6)	34 (94.4)	<0.001
Mediastinitis; *n* (%)	4 (22.2)	10 (27.8)	0.751
Steroids; *n* (%)	2 (11.1)	13 (36.1)	0.062
COPD; *n* (%)	2 (11.1)	16 (44.4)	0.016
Respirator days; mean (SD)	13.3 (12.4)	12.2 (11.7)	0.713
ICU days; mean (SD)	17.9 (14.4)	16.5 (14.3)	0.582
Adjusted mortality; *n* (%)	2 (11.1)	5 (13.9)	1
All-cause mortality; *n* (%)	8 (44.4)	19 (52.8)	0.773

Intensive care unit (ICU); chronic obstructive pulmonary disease (COPD); number (*n*); standard deviation (SD); *p*-value (*p*). Patients receiving surgical repair had higher rates of emphysema (*p* < 0.001), interfacility referral (*p* = 0.014), and COPD (*p* = 0.016) compared to patients without surgical repair. Adjusted and all-cause mortality were comparable (*p* = 0.773 and *p* = 1).

**Table 3 jcm-09-00382-t003:** Univariate analysis for survival of iatrogenic tracheal rupture.

	All-Cause Mortality		Adjusted Mortality	
Term	HR (95% CI)	*p*	HR (95% CI)	*p*
Age	1 (0.99–1)	0.222	1 (0.98–1.1)	0.276
COPD	1.8 (0.81–3.8)	0.151	3.4 (0.75–15)	0.114
Emphysema	8.3 (1.1–62)	0.038	n.a.*	n.a.*
Female	0.75 (0.32–1.8)	0.521	0.7 (0.14–3.6)	0.673
Height	1 (0.96–1.1)	0.643	1 (0.92–1.2)	0.639
Mediastinitis	2.3 (1.1–5)	0.035	8.2 (1.6–43)	0.012
Rupture length	1.6 (1.3–2)	<0.001	1.7 (1.1–2.7)	0.019
Steroids	1.6 (0.71–3.5)	0.262	4.2 (0.93–19)	0.063
Surgical repair	1.3 (0.55–2.9)	0.577	1.3 (0.26–6.8)	0.742

Hazard ratio (HR); confidence interval (CI); chronic obstructive pulmonary disease (COPD); *p*-value (*p*); not applicable (n.a.). Effects were seen for both outcome-definitions for mediastinitis and rupture lengths and additionally for all-cause-mortality for emphysema (*n* = 54; 27 events for all-cause mortality and seven events for adjusted mortality). * Estimation not possible due to limited number of observations.

**Table 4 jcm-09-00382-t004:** Multivariate analysis for survival of iatrogenic tracheal rupture.

	All-Cause Mortality		Adjusted Mortality	
Term	HR (95% CI)	*p*	HR (95% CI)	*p*
Emphysema	3.0 (0.4–25)	0.301	-	-
Mediastinitis	1.6 (0.7–3.5)	0.243	5.8 (1.1–31.7)	0.042
Rupture length	1.5 (1.2–1.9)	0.001	1.5 (0.97–2.3)	0.068

Hazard ratio (HR); confidence interval (CI); *p*-value (*p*). Covariates in the multivariate analysis included significant variables in the univariate analysis in Table 3. Emphysema, mediastinitis, and rupture length were adjusted for all-cause mortality; mediastinitis and rupture length were adjusted for adjusted mortality.

**Table 5 jcm-09-00382-t005:** Comparison of characteristics between rupture length <4.5 cm and ≥4.5 cm.

Parameter	<4.5 cm	≥4.5 cm	*p*
Patients; *n*	30	24	
Female; *n* (%)	26 (86.7)	14 (58.3)	0.028
Age, years; mean (SD)	64.7 (16.9)	69.6 (15.4)	0.216
Weight, kg; mean (SD)	75.5 (16.2)	76.5 (22.0)	0.951
Height, cm; mean (SD)	164.0 (6.4)	165.5 (7.1)	0.421
ICU prior rupture; *n* (%)	17 (56.7)	12 (50.0)	0.784
Interfacility referral; *n* (%)	8 (26.7)	11 (45.8)	0.164
Rupture length, cm; mean (SD)	3.0 (1.0)	6.0 (1.0)	<0.001
Emphysema; *n* (%)	20 (66.7)	24 (100)	0.001
Mediastinitis; *n* (%)	6 (20.0)	8 (33.3)	0.353
Steroids; *n* (%)	6 (20.0)	9 (37.5)	0.223
COPD; *n* (%)	8 (26.7)	10 (41.7)	0.264
Surgical repair; *n* (%)	16 (53.3)	20 (83.3)	0.024
Respirator days; mean (SD)	14.2 (12.1)	10.5 (11.4)	0.236
ICU days; mean (SD)	19.9 (13.6)	13.3 (14.4)	0.028
Adjusted mortality; *n* (%)	1 (3.3)	6 (25.0)	0.036
All-cause mortality; *n* (%)	8 (26.7)	19 (79.2)	<0.001

Intensive care unit (ICU); chronic obstructive pulmonary disease (COPD); number (*n*); standard deviation (SD); p-value (*p*). A rupture size of ≥4.5 cm was associated with higher rates of emphysema (*p* = 0.001), higher rates of surgical repair (*p* = 0.024), male sex (*p* = 0.028), and fewer ICU days (*p* = 0.028) than a rupture size of <4.5 cm.

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
