# Peer review of "Prognostic Factors for Iatrogenic Tracheal Rupture: A Single-Center Retrospective Cohort Study"

_jcm, 2020, doi:10.3390/jcm9020382_

Round 1

Reviewer 1 Report

The authors present a retrospective chart review prognostic determinants of iatrogenic tracheal rupture. A total of 54 patients were ultimately evaluated. Rupture length was identified as the single prognostic factor for all-cause mortality on univariate and multivariate analysis and the presence of mediastinitis was predictive of adjusted mortality. The authors conclude that these two elements of tracheal rupture may be predictors of mortality as described. The paper is well-written and easy to read, and appears to be a worthwhile contribution to the current literature.

Comments:

I would like to better understand the pathology being evaluated. You mentioned that several patients had tracheal rupture associated with tracheostomy. How were the tracheostomy and tracheal rupture related? Were these separate perforations (the tracheostomy and the rupture) or one lesion in continuity (tracheal tear at trach site)? Depending on the answer above, it would be interesting to understand if the location (rather than just the length) was predictive of survival; likewise, the time between identification and repair may be influential on survival in isolation; lastly, a history of prior or concurrent thoracic surgery may influence the severity of tracheal rupture, as lesions which are contained leaks may be less emergent and/or problematic than those which promote free extravasation of air and secretions.

Author Response

Point-by-point response to Reviewer´s comments

Reviewer 1

The authors present a retrospective chart review prognostic determinants of iatrogenic tracheal rupture. A total of 54 patients were ultimately evaluated. Rupture length was identified as the single prognostic factor for all-cause mortality on univariate and multivariate analysis and the presence of mediastinitis was predictive of adjusted mortality. The authors conclude that these two elements of tracheal rupture may be predictors of mortality as described. The paper is well-written and easy to read, and appears to be a worthwhile contribution to the current literature.

Comments:

I would like to better understand the pathology being evaluated. You mentioned that several patients had tracheal rupture associated with tracheostomy. How were the tracheostomy and tracheal rupture related? Were these separate perforations (the tracheostomy and the rupture) or one lesion in continuity (tracheal tear at trach site)? Depending on the answer above, it would be interesting to understand if the location (rather than just the length) was predictive of survival;

RESPONSE: All lesions of tracheotomy-related ruptures were opposite to the tracheotomy site. Patients with tracheotomy-related tracheal ruptures had comparable survival rates with tracheal intubation-related ruptures and surgery-related ruptures (Figure 1). We have rewritten the section 2.1 Study Patients, Design and Definitions to clarify that all iatrogenic tracheal ruptures were at the pars membranacea, the posterior part of the trachea. Furthermore, we have added in the Discussion section (4.1.):  “Different causative procedures may be associated with different patterns of injuries.” and “For tracheotomy-related ruptures and surgery-related ruptures, some of these risk factors may also be applicable but detailed data are not available. In our study, all tracheotomy-related ruptures were opposite to the tracheotomy site at the posterior wall of the trachea.”

likewise, the time between identification and repair may be influential on survival in isolation;

RESPONSE: We agree with the reviewer that the time between identification and treatment may be outcome-relevant. In our study, all iatrogenic tracheal ruptures were identified and treated within one day apart from five patients who remained asymptomatic until tracheotomy/bronchoscopy (3.2 Baseline Characteristics of Patients). We have added to the Results section: “All other patients were identified and treated within one day after rupture.”

lastly, a history of prior or concurrent thoracic surgery may influence the severity of tracheal rupture, as lesions which are contained leaks may be less emergent and/or problematic than those which promote free extravasation of air and secretions.

RESPONSE: We agree with the reviewer, however, only a small number (11%) of our patients had surgery-related tracheal rupture. As mentioned above, we have added in the Discussion section (4.1.): “For tracheotomy-related ruptures and surgery-related ruptures, some of these risk factors may also be applicable but detailed data are not available. In our study, all tracheotomy-related ruptures were opposite to the tracheotomy site at the posterior wall of the trachea. Furthermore, the role of prior thoracic surgery on the severity of iatrogenic rupture and their outcomes was difficult to assess and should be evaluated in future studies.”

We would like to thank the reviewer for his/her constructive comments and hope that we have addressed all items appropriately.

We have changed some minor issues in Figures 1 and 3 (missing letters, p values in italics) and provided significance levels (0.xxx instead of 0.xx) in the main text according to Table 3.

The manuscript has been processed by a professional language and style editing (AJE).

We have changed some words according to iThenticate matches, as suggested by the Editor.

Furthermore, we have now added our raw data file (Table S1).

We are open for any further comments, thank you again.

Reviewer 2 Report

I would like to thank the authors for this opportunity to review their work. In this large single center retrospective series the authors describe risk factors for mortality in patients who suffer from iatrogenic tracheal rupture. The length and depth of the tracheal injury have been known to be associated with increased mortality. Perhaps, the authors could describe why they chose 4.5 cm as a cutoff for the length of the injury. 
Could the authors please describe how patients were chosen for surgical versus conservative management?

The authors describe that 81 per cent of their patients had subcutaneous emphysema. Patients with iatrogenic tracheal injuries may often remain asymptomatic. If possible please include the number of patients who remained asymptomatic. 

Could you please explain why instead of describing an odds ratio for univariate analysis why was an ROC analysis performed? 

Author Response

Point-by-point response to Reviewer´s comments

Reviewer 2

I would like to thank the authors for this opportunity to review their work. In this large single center retrospective series the authors describe risk factors for mortality in patients who suffer from iatrogenic tracheal rupture. The length and depth of the tracheal injury have been known to be associated with increased mortality. Perhaps, the authors could describe why they chose 4.5 cm as a cutoff for the length of the injury. 

RESPONSE: The cutoff length 4.5 cm was the result of ROC curve analysis (Figure 2) for best match of the Youden index. We have added in the Results section (3.4.): “We chose the cutoff length at the best Youden index, as this point provided the highest sum of sensitivity and specificity to predict all-cause mortality”.

Could the authors please describe how patients were chosen for surgical versus conservative management?

RESPONSE: We have described the decision process for or against surgery in the Methods section (2.2 General Management) and discussed this point in 4.3 Surgical and Non-Surgical Treatment. For clarity, we have added in the Methods section (2.2.): “All decisions were based on an individual approach of weighting risks and benefits of surgical repair or conservative treatment and did not follow a study protocol.” As this is an important point we hope that we have presented our data clear enough.

The authors describe that 81 per cent of their patients had subcutaneous emphysema. Patients with iatrogenic tracheal injuries may often remain asymptomatic. If possible please include the number of patients who remained asymptomatic. 

RESPONSE: We have added in the results section: “Subcutaneous emphysema was present in 44 patients (81%) as primary clinical symptom, while nine patients (19%) remained asymptomatic.“

Could you please explain why instead of describing an odds ratio for univariate analysis why was an ROC analysis performed?

RESPONSE: ROC curve analysis was performed in order to calculate a cutoff rupture length using the Youden index. Univariate risk analysis of rupture length is provided in Table 3, where we report that an increase of 1 cm rupture length is associated with an HR for all-cause mortality of 1.6 (95% CI 1.3-2.0; p < 0.001). We have added our explanation in 3.4. Analysis of Rupture Length and Mortality: “We chose the cutoff length at the best Youden index, as this point provided the highest sum of sensitivity and specificity to predict all-cause mortality.”

We would like to thank the reviewer for his/her constructive comments and hope that we have addressed all items appropriately.

We have changed some minor issues in Figures 1 and 3 (missing letters, p values in italics) and provided significance levels (0.xxx instead of 0.xx) in the main text according to Table 3.

The manuscript has been processed by a professional language and style editing (AJE).

We have changed some words according to iThenticate matches, as suggested.

Furthermore, we have now added our raw data file (Table S1).

We are open for any further comments, thank you again.